# Removal of Toxic Heavy Metals from Contaminated Aqueous Solutions Using Seaweeds: A Review

Edward Hingha Foday Jr [1,2,3], Bai Bo [1,2,4,5,*] and Xiaohui Xu [1,2]

1    Key Laboratory of Subsurface Hydrology and Ecological Effects in Arid Region of the Ministry of Education,
     Chang'an University, Xi'an 710054, China; hinghaja@gmail.com (E.H.F.J.); xx4@princeton.edu (X.X.)
2    Department of Environmental Engineering, School of Water and Environment, Chang'an University,
     Xi'an 710054, China
3    Faculty of Education, Eastern Technical University of Sierra Leone, Combema Road,
     Kenema City 00232, Sierra Leone
4    Key Laboratory of Tibetan Medicine Research, Northwest Institute of Plateau Biology,
     Chinese Academy of Sciences, Xining 810008, China
5    Qinghai Provincial Key Laboratory of Tibetan Medicine Research, Xining 810001, China
*    Correspondence: baibochina@163.com

**Abstract:** Heavy metal contamination affects lives with concomitant environmental pollution, and seaweed has emerged as a remedy with the ability to save the ecosystem, due to its eco-friendliness, affordability, availability, and effective metal ion removal rate. Heavy metals are intrinsic toxicants that are known to induce damage to multiple organs, especially when subjected to excess exposure. With respect to these growing concerns, this review presents the preferred sorption material among the many natural sorption materials. The use of seaweeds to treat contaminated solutions has demonstrated outstanding results when compared to other materials. The sorption of metal ions using dead seaweed biomass offers a comparative advantage over other natural sorption materials. This article summarizes the impact of heavy metals on the environment, and why dead seaweed biomass is regarded as the leading remediation material among the available materials. This article also showcases the biosorption mechanism of dead seaweed biomass and its effectiveness as a useful, cheap, and affordable bioremediation material.

**Keywords:** heavy metals; seaweed; biosorption; aqueous solution; remediation

## 1. Introduction

The severity of heavy metal pollution cannot be over-emphasized, as it has become a universal issue in recent years. The effects of heavy metals in the environment are harmful due to their high toxicity. Their release into the environment occurs as a result of various natural and anthropogenic activities. Unfortunately, most of these heavy metals, whether generated from human activities or nature, constantly undermine the existence and health of environmental resources. The toxicity, persistence, and non-biodegradable nature of these metal ions make them a threat to the environment [1,2]. These heavy metals are known to cause multiple and complicated health problems such as brain and lung damage, cancer, nausea, and vomiting [3,4]. Seaweed, also known as marine algae, serves as one of the major leading biosorption materials for the treatment of heavy metals [5]. Seaweed produces a variety of compounds such as xanthophylls, chlorophyll, carotenoids, vitamins, fatty acids, amino acids as well as antioxidants (such as halogenated compounds, alkaloids, and polyphenols), and polysaccharides (such as agar, alginate, carrageenan, proteoglycans, galactosyl glycerol, laminarin, rhamnan sulfate, and fucoidan) [6]. The presence of alginate in the seaweed makes it an effective eluted material for metal ion removal. Alginate, as well as fucoidan, has a high sorption capacity, which can mainly be attributed to polysaccharides found in the cell walls. The carboxylic and sulfonic acid functional groups are more active in the ion exchange process, and polysaccharides are

responsible for these functional groups [5,7]. On the whole, seaweed has proven to be one of the most outstanding and important biosorption materials for the remediation of metal ions. Its low cost, availability, and eco-friendliness, coupled with its high metal ion uptake capability, make it an ideal biosorption material compared to other sorption materials [6,8]. In this review, dead seaweed biomass is of particular interest, and because of the scant knowledge regarding its usefulness and biosorption mechanism, we seek to throw light on the importance of dead seaweed biomass as a sorption material and to summarize its biosorption mechanism. This review also pinpoints the toxic effects of heavy metals on environmental resources, as well as comparing dead seaweed biomass with other natural sorption materials in terms of heavy metal removal.

## 2. Heavy Metal Contamination in Water

Water is a universal solvent needed by all living organisms and is also good at dissolving both organic and inorganic compounds. Water resources are critically affected by heavy metal contamination, and this has seriously altered the aquatic ecosystem [9]. On a large scale, aquatic ecosystems are contaminated by heavy metals from industrial effluent, domestic sewage, and agricultural runoff [10]. Most rivers, streams, and lakes are polluted through erosion and leaching, while atmospheric deposition, metal corrosion, sediment resuspension, and metal evaporation are some of the ways the environment gets polluted [11,12]. The non-biodegradable character of heavy metals and their persistence in the environment have led to bioaccumulation through the food chain, leading to complicated health issues and environmental pollution [13]. The term heavy metals refer to metals and metalloids whose mass is over 5 g/cubic centimeters (g/cm$^3$) and are naturally occurring elements commonly found on earth [14]. They can be regarded as trace elements due to their trace concentrations in the environment. The set of environmental matrices for metal ion concentrations range from zero (0) ppb to ten (10) ppb [15,16]. Anthropogenic and natural activities such as mining, fossil fuel combustion, agriculture, volcanic eruptions, earthquakes, weathering of rocks, and industrial activity are the main causes of environmental contamination [17]. Direct contact with these heavy metals either through inhalation or ingestion poses serious health threats such as teratogenesis, cancer, and internal disorders [18]. Cadmium (Cd), Chromium (Cr), Lead (Pb), Mercury (Hg), and Arsenic (As) were identified by Tchounwou and team [16] as the most toxic heavy metals, and have been placed under the category "priority metals", which means they are metals of public concern, due to their toxic nature. These aforementioned metal ions are innately toxic and are capable of inducing damage to multiple organs even at minimal exposure levels. Reactive oxygen species (ROS) together with oxidative stress (OS) play key roles in the carcinogenic and toxic nature of these metal ions [16]. Zinc (Zn), Copper (Cu), Molybdenum (Mo), and several other metals have also been considered essential elements because they assist in biochemical reactions, although excess exposure above the required threshold can impair human health [19]. Against this background, international institutions like the United States Environmental Protection Agency (USEPA), the World Health Organization (WHO), the European Union (EU), etc. have set acceptable thresholds referred to as Maximum Contaminant Levels (MCLs). Table 1 shows the internationally accepted thresholds of metal ion concentrations in drinking water.

**Table 1.** Accepted thresholds of toxic metal ions in drinking water.

| Metals | WHO [20] | USEPA [21] | EU Standard [22] | MEE-China [23] | DWI-UK [24] |
|---|---|---|---|---|---|
| | Drinking Water Acceptable Standards in (mg L$^{-1}$) | | | | |
| Nickel (Ni) | 0.07 | - | 0.020 | 0.000 | 0.02 |
| Lead (Pb) | 0.01 | 0.015 | 0.005 | 0.010 | 0.01 |
| Zinc (Zn) | - | 5.0 | - | 0.05 | - |
| Copper (Cu) | 2.0 | 1.0 | 2.000 | 1.000 | 2.0 |
| Cadmium (Cd) | 0.003 | 0.005 | 0.005 | 0.005 | 0.005 |
| Mercury (Hg) | 0.006 | 0.002 | 0.001 | 0.00005 | 0.001 |
| Arsenic (As) | 0.01 | 0.010 | 0.01 | 0.050 | 0.01 |
| Chromium (Cr) | 0.05 | 0.100 | 0.025 | 0.050 | 0.05 |
| Antimony | 0.02 | - | 0.01 | - | 0.005 |
| Bromate | 0.01 | - | 0.01 | - | 0.01 |
| Uranium | 0.03 | 0.03 | 0.03 | - | - |

The contamination of water bodies normally happens through leaching, erosion, wind, and other environmental means, thereby leading to negative health implications and risk to the ecosystem. Heavy metal pollution leaves a negative blueprint on the environment and people's lives. As shown in Figures 1 and 2, natural and anthropogenic sources are the known sources for heavy metal contamination. The natural sources for these toxic metals include volcanic eruptions, forest fires, biogenic sources, and the weathering of rock [25], while industrial estates, automobile exhaust, the spraying of insecticide, agricultural activities, transportation, and mining are the main anthropogenic sources of heavy metals pollution [26].

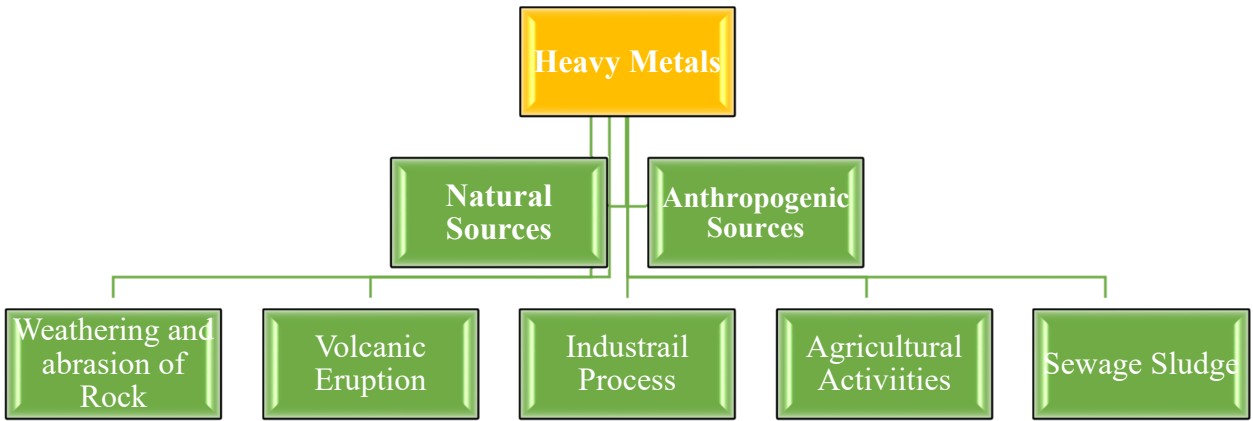

**Figure 1.** Categories of heavy metal sources.

As seen in Figure 3 below, topsoil and underground water are normally polluted by industrial activities, agricultural activities, weathering, volcanic eruptions, and other biogenic activities. The water bodies become contaminated as the topsoil is washed into them by either erosion, leaching, or landfill leakage. In turn, flora and fauna are affected as the polluted water bodies are consumed and accumulated into their systems, tissues, and organs. Human beings, on the receiving end, are exposed to multiple risks of biochemical disorder or organ failures following the ingestion of contaminated plants and animals.

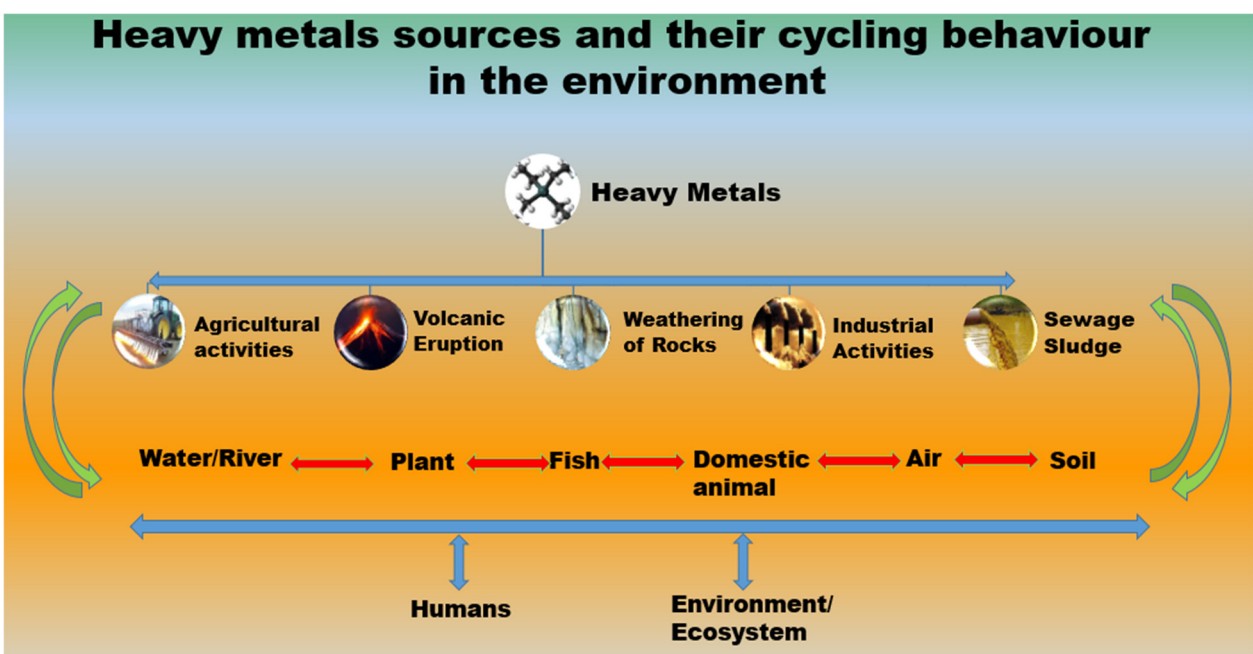

**Figure 2.** Sources of heavy metals.

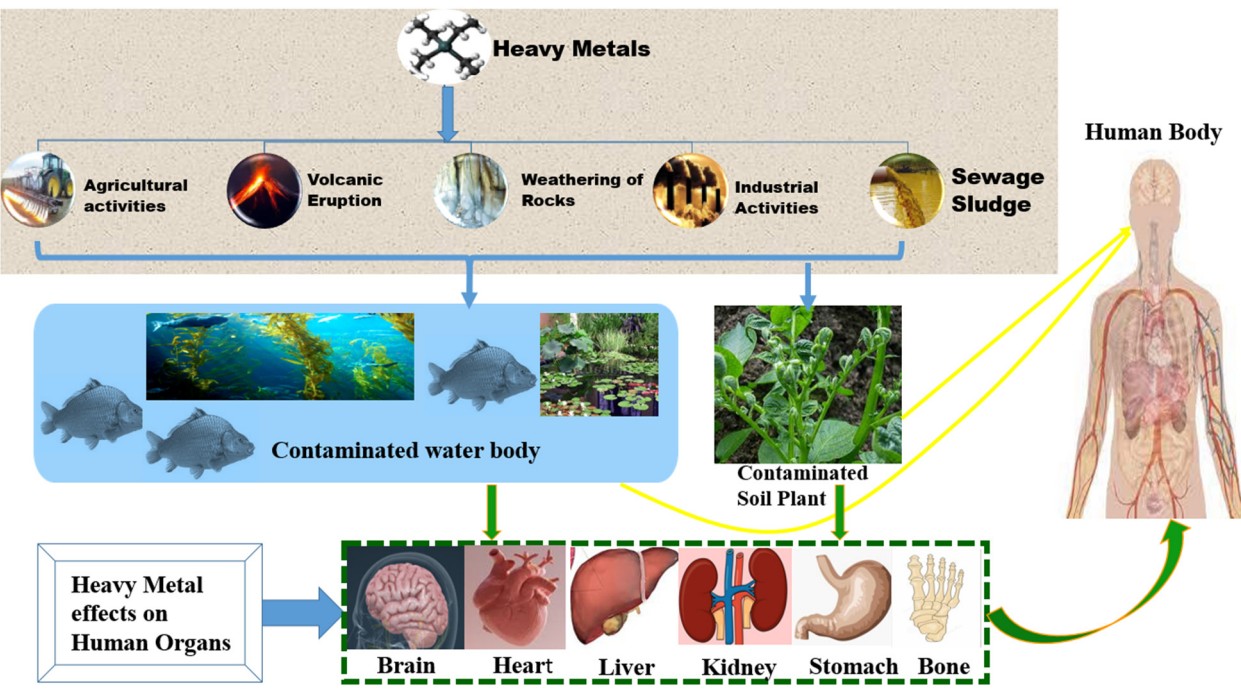

**Figure 3.** Heavy metal contamination in water.

## 3. Structure and Classification of Seaweed

Seaweed does not have roots, but rather has holdfasts that anchor the seaweed to the bottom of the sea or ocean. These root-like holdfasts are composed of many finger-like components known as Haptera and are supported by a stalk or stem called a Stipe. The structure of the stem or stipe can be hard, filled with gas, soft or flexible, short, or long, and in some cases, they may be completely absent depending on the type of seaweed [27]. These stipes or stem-like structures are either filled with gas or empty. These are referred to as pneumatocysts, while the entire body of the seaweed is referred to as the thallus. Seaweed has leaves called blades, which assist in photosynthesis, although some seaweed

species have only a single leaf, while others have many leaves. Figure 4 below shows the physical structure of seaweed.

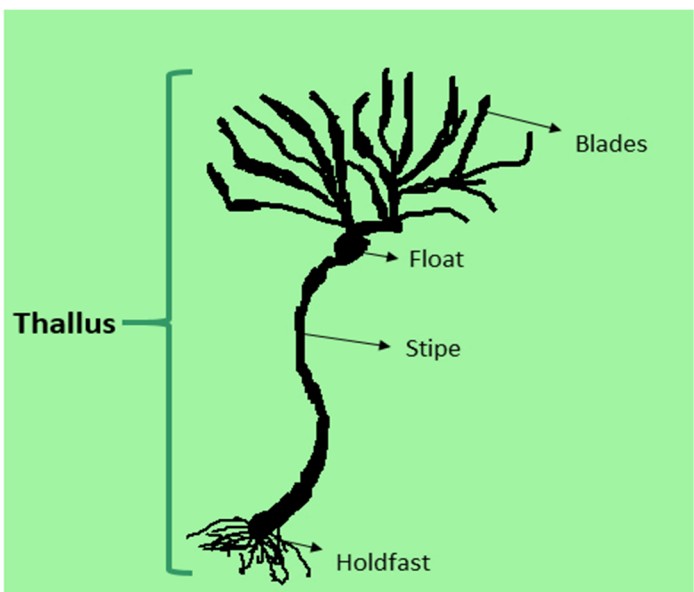

**Figure 4.** Structure of seaweed.

Seaweed is divided into three (3) main groups based on color characterization, namely: Brown (Phaeophyceae), Red (Rhodophyceae), and Green (Chlorophyceae) seaweeds [28]. Brown algae (Phaeophyta) have various physical appearances either in crust or filament form. Brown algae are multicellular and contain chlorophyll, which aids in photosynthesis, with fucoxanthin being the dominant pigment. Physically, brown algae can range from a large size (Kelp) of about 60 m long to as small as 60 cm [29]. Red algae (Rhodophyta) have chlorophyll in which phycocyanin and phycoerythrin are the dominant pigments responsible for red coloration. Red seaweeds are normally not actually red, but brownish-red or purple. Physically, red algae are smaller than brown algae in length [30]. Green seaweeds (chlorophyte) have chlorophyll, but with no dominant pigment justifying their green coloration; therefore, green seaweed is generally green. It is smaller in size than both red and brown seaweeds [5,31].

We further characterized seaweeds based on both their physical and chemical compositions as shown in Table 2. The alginate and the intercellular substance of the brown algae have high divalent cation uptakes. The cell walls of brown seaweeds are composed of cellulose, alginic acid, and polysaccharides, with alginates and sulfate being the dominant active groups [7]. The cell wall of red algae contains cellulose, but their biosorption capabilities can largely be attributed to sulfated polysaccharides made up of galactans. Similarly, the cell wall of the green algae contains cellulose with hydroxyl-proline glucosides; xylans and mannans are the main functional groups during biosorption [32,33].

**Table 2.** Characteristics of Seaweed.

| Common Name (Phylum) | Body Form | Size | Pigments | Colour Composition | Cell Walls |
|---|---|---|---|---|---|
| Brown algae (Phaeophyta) | Multicellular | 60 cm–60 m | Chlorophyll, Fucoxanthin, and several other xanthophylls | Golden-brown, Greenish-brown | Cellulose, Alginate, Fucoidan |
| Red algae (Rhodophyta) | Multicellular | 50 cm–2 m | Chlorophyll, Phycocyanin, Phycoerythrin, and several xanthophylls | Brownish red, Purple | Cellulose, Xylans, Galactans |
| Green algae (Chlorophyta) | Unicellular, Colonial, Filamentous, Multicellular | 1–1000 μm | a and b Chlorophyll and several xanthophylls | Green | Cellulose Hydroxyl –proline glucosides β- xylans, β-mannans |

### 3.1. Seaweed: Metal Ion Biosorption Material

The treatment of contaminated solutions has been a burden to engineers and scientists over the years. Recently, seaweed has been proven to be more effective than other natural sorption materials. Some of the other natural sorption materials that have been used to elute metal ions are discussed in the next subsection. Remediation of aqueous solution from metal ions is of serious concern to environmentalists, considering the threat it poses to the purity of the natural environment [34]. The non-biodegradability, carcinogenicity, and toxicity of heavy metals make them harmful, and treatment of these heavy metals is essential [35]. Sorption has been proven to be a sustainable and effective method for treating heavy metals in aqueous solutions using natural biomass [36]. Based on these outstanding results, seaweed has emerged as the leading material, with a high rate of metal ion removal. The biosorption method is one of the simplest, cheapest, and most eco-friendly methods, and requires little or no nutrient addition. The effectiveness and efficiency of treatments for heavy metals are directly related to the type of sorbent used [37]. In short, the remediation of heavy metals using seaweed offers a more reliable, cheaper, and more effective means of heavy metal removal from aqueous solutions than the previous methods. Various mechanisms of seaweed biomass (electrostatic interaction, ion exchange, and complex formation) have been used in the biosorption process of heavy metals, and ion exchange has been widely used and is considered the most important among the list of mechanisms [38,39]. The cell walls of the algae possess polysaccharides and protein, which serve as binding sites for metal ion uptake [40]. There are several factors responsible for the sorption capability of a seaweed cell surface; among these factors are accessibility of binding groups for metal ions, the affinity constants of the metal with the functional group, the chemical state of these sites, the number of functional groups in the algae matrix, and the coordination number of the metal ion to be sorbed [41]. The metal biosorption ability of seaweed varies because of the heterogeneity of their respective cell wall composition. For example, as seen in Table 3, brown, green, and red algae have high affinities for lead (Pb), copper (Cu), and cobalt (Co), respectively [7]. Physical or chemical treatment can enhance heavy metal uptake by seaweed, and the cell wall surface is modified, thereby providing additional binding sites for biosorption [7,42]. The physical treatment includes freezing, crushing, heating, and drying, as these increase the surface area on which biosorption can be achieved [42]. The most common seaweed pretreatments are glutaraldehyde, calcium-chloride ($CaCl_2$), formaldehyde, sodium hydroxide (NaOH), and hydrogen-chloride (HCl). Pretreatment with calcium-chloride (CaCl2) enhances calcium binding with alginate, which plays a pivotal role in ion exchange [43]. The crosslinking bond between hydroxyl and amino group is strengthened by formaldehyde and glutaraldehyde [44]. The electrostatic interactions of metal ion cations are increased by sodium hydroxide (NaOH), while at the same time providing optimal conditions for ion exchange, while hydrogen-chloride

(HCl) dissolves the polysaccharides of the cell wall and also replaces light metal ions with a proton, thereby increasing the biosorption binding sites [7]. It is in this regard that we aim to showcase the comparative advantages of seaweed over other sorption materials in the removal of heavy metals.

**Table 3.** Different algae species for heavy metal removal.

| Species of Algae | Metal Ions | qmax (mmol/g) | pH | References |
|---|---|---|---|---|
| **Green Algae** | | | | |
| *Ulva lactuca* | | 0.61 | 4.5 | [45] |
| *Cladophora glomerata* | | 0.35 | 4.5 | [45] |
| *Ulva* sp. | | 1.46 | 5.0 | [33] |
| *Codium vermilara* | | 0.30 | 5.0 | [46] |
| *Spirogyra insignis* | Pb(II) | 0.24 | 5.0 | [46] |
| *Spirogyra neglecta* | | 0.56 | 5.0 | [47] |
| *Caulerpa lentillifera* | | 0.13 | 5.0 | [48] |
| *Spirogyra* sp. | | 0.43 | 5.0 | [49] |
| *Cladophora* sp. | | 0.22 | 5.0 | [49] |
| *Ulva* sp. | | 0.75 | 5.0 | [33] |
| *Codium vermilara* | | 0.26 | 5.0 | [46] |
| *Spirogyra insignis* | | 0.30 | 4.0 | [46] |
| *Spirogyra neglecta* | Cu(II) | 1.80 | 4.5 | [47] |
| *Ulva fasciata* | | 1.14 | 5.5 | [50] |
| *Caulerpa lentillifera* | | 0.08 | 5.0 | [48] |
| *Cladophora sp* | | 0.23 | 5.0 | [49] |
| *Spirogyra sp* | | 0.53 | 5.0 | [51] |
| *Ulva* sp. | | 0.58 | 5.5 | [33] |
| *Chaetomorpha linum* | | 0.48 | 5.0 | [52] |
| *Codium vermilara* | | 0.19 | 6.0 | [46] |
| *Spirogyra insignis* | Cd(II) | 0.20 | 6.0 | [46] |
| *Ulva lactuca* | | 0.25 | 5.0 | [53] |
| *Oedogonium* sp. | | 0.79 | 5.0 | [54] |
| *Caulerpa lentillifera* | | 0.04 | 5.0 | [48] |
| *Spirogyra* sp. | | 0.006 a | - | [55] |
| *Ulva* sp. | | 0.54 | 5.5 | [33] |
| *Codium vermilara* | | 0.36 | 6 | [46] |
| *Spirogyra insignis* | Zn(II) | 0.32 | 6 | [46] |
| *Caulerpa lentillifera* | | 0.04 | 5 | [48] |
| *Spirogyra s* | | 0.02 a | - | [55] |
| *Ulva* sp. | | 0.29 | 5.5 | [33] |
| *Codium vermilara* | Ni(II) | 0.22 | 6.0 | [46] |
| *Spirogyra insignis* | | 0.29 | 6.0 | [46] |
| *Ulva lactuca* | | 1.14 | 4.5 | [56] |
| **Red Algae** | | | | |
| *Gracilaria corticata* | | 0.26 | 4.5 | [45] |
| *Gracilaria canaliculata* | | 0.20 | 4.5 | [45] |
| *Polysiphonia violacea* | | 0.49 | 4.5 | [45] |
| *Gracillaria* sp. | | 0.45 | 5.0 | [33] |
| *Asparagopsis armata* | | 0.30 | 4.0 | [46] |
| *Jania rubens* | | 0.14 | 5.0 | [57] |
| *Pterocladia capillacea* | | 0.16 | 5.0 | [57] |
| *Corallina mediterranea* | | 0.31 | 5.0 | [57] |
| *Galaxaura oblongata* | | 0.42 | 5.0 | [57] |
| *Asparagopsis armata* | | 0.33 | 5.0 | [46] |
| *Chondrus crispus* | Pb(II) | 0.63 | 4.0 | [46] |
| *Gelidium* | | 0.51 | 5.3 | [58] |
| *Gracilaria changii* | | 0.23 | 5.0 | [52] |
| *Gracilaria edulis* | | 0.24 | 5.0 | [52] |

**Table 3.** *Cont.*

| Species of Algae | Metal Ions | qmax (mmol/g) | pH | References |
|---|---|---|---|---|
| *Gracilaria Salicornia* | | 0.16 | 5.0 | [52] |
| *Asparagopsis armata* | | 0.28 | 6.0 | [46] |
| *Ceramium virgatum* | | 0.35 | 5.0 | [59] |
| *Mastocarpus stellatus* | | 0.59 | 6.0 | [60] |
| *Jania rubens* | | 0.27 | 5.0 | [57] |
| *Corallina mediterranea* | | 0.57 | 5.0 | [57] |
| *Hypnea valentiae* | | 0.15 | 6.0 | [61] |
| *Palmaria palmate* | | 0.57 (Cr(III)) | 4.5 (Cr(III | [62] |
| | | 0.65 (Cr(VI)) | 2 (Cr(VI)) | |
| *Polysiphonia lanosa* | | 0.65 (Cr(III)) | 4.5(Cr(III)) | [62] |
| | | 0.88 (Cr(VI)) | 2 (Cr(VI)) | |
| *Jania rubens* | Cr | 0.54 (Cr(III)) | 5.0 (Cr(III)) | [57] |
| *Pterocladia capillacea* | | 0.66 (Cr(III)) | 5.0 (Cr(III)) | [57] |
| *Corallina mediterranea* | | 1.35 (Cr(III)) | 5.0 (Cr(III)) | [57] |
| *Galaxaura oblongata* | | 2.02 (Cr(III)) | 5.0 (Cr(III)) | [57] |
| *Jania rubens* | | 0.55 | 5.0 | [57] |
| *Pterocladia capillacea* | Co(II) | 0.89 | 5.0 | [57] |
| *Corallina mediterranea* | | 1.29 | 5.0 | [57] |
| *Galaxaura oblongata* | | 1.25 | 5.0 | [57] |
| **Brown Algae** | | | | |
| *Ascophyllum nodosum* | | 1.31 | 3.5 | [63] |
| *Fucus vesiculosus* | | 1.11 | 3.5 | [63] |
| *Sargassum vulgare* | | 1.10 | 3.5 | [63] |
| *Sargassum hystrix* | | 1.37 | 4.5 | [45] |
| *Sargassum natans* | | 1.14 | 4.5 | [45] |
| *Padina pavonia* | Pb(II) | 1.04 | 4.5 | [45] |
| *Sargassum* sp. | | 1.16 | 5.0 | [33] |
| *Padina* sp. | | 1.25 | 5.0 | [33] |
| *Fucus vesiculosus* | | 1.02 | 5.0 | [38] |
| *Fucus spiralis* | | 0.98 | 3.0 | [46] |
| *Ascophyllum nodosu* | | 0.86 | 3.0 | [46] |
| *Padina* sp. | | 1.14 | 5.0 | [33] |
| *Sargassum vulgarie* | | 0.93 | 4.5 | [64] |
| *Sargassum fluitans* | | 0.80 | 4.5 | [64] |
| *Sargassum filipendula* | | 0.89 | 4.5 | [64] |
| *Fucus vesiculosus* | | 1.66 | 5.0 | [38] |
| *Fucus spiralis* | Cu(II) | 1.10 | 4.0 | [46] |
| *Ascophyllum nodosum* | | 0.91 | 4.0 | [46] |
| *Sargassum filipendula* | | 1.32 | 4.5 | [65] |
| *Fucus serratus* | | 1.60 | 5.5 | [66] |
| *Sargassum* sp. | | 1.13 | 5.5 | [50] |
| *Sargassum* sp. | | 0.76 | 5.5 | [33] |
| *Padina sp* | | 0.75 | 5.5 | [33] |
| *Sargassum siliquosum* | | 0.73 | 5.0 | [52] |
| *Sargassum baccularia* | | 0.74 | 5.0 | [52] |
| *Padina tetrastomatica* | | 0.53 | 5.0 | [52] |
| *Sargassum vulgarie* | | 0.79 | 4.5 | [64] |
| *Sargassum fluitans* | | 0.71 | 4.5 | [64] |
| *Sargassum muticum* | | 0.68 | 4.5 | [64] |
| *Fucus vesiculosus* | | 0.96 | 6.0 | [38] |
| *Fucus spiralis* | Cd(II) | 1.02 | 6.0 | [46] |
| *Ascophyllum nodosum* | | 0.78 | 6.0 | [46] |
| *Sargassum filipendula* | | 1.17 | 5.0 | [67] |
| *Bifurcaria bifurcate* | | 0.65 | 4.5 | [68] |
| *Saccorhiza polyschides* | | 0.84 | 4.5 | [68] |

**Table 3.** *Cont.*

| Species of Algae | Metal Ions | qmax (mmol/g) | pH | References |
|---|---|---|---|---|
| *Ascophyllum nodosum* | | 0.70 | 4.5 | [68] |
| *Laminaria ochroleuca* | | 0.56 | 4.5 | [68] |
| *Pelvetia caniculata* | | 0.66 | 4.5 | [68] |
| *Macrocystis pyrifera* | | 0.89 | 3.0 | [69] |
| *Sargassum* sp. | | 0.50 | 5.5 | [33] |
| *Padina* sp. | | 0.81 | 5.5 | [33] |
| *Fucus spiralis* | Zn(II) | 0.81 | 6.0 | [46] |
| *Ascophyllum nodosum* | | 0.64 | 6.0 | [46] |
| *Sargassum filipendula* | | 0.71 | 5.0 | [67] |
| *Macrocystis pyrifera* | | 0.91 | 4.0 | [69] |
| *Sargassum fluitans* | | 0.75 | 3.5 | [63] |
| *Ascophyllum nodosum* | | 0.69 | 3.5 | [63] |
| *Sargassum natans* | | 0.41 | 3.5 | [63] |
| *Fucus vesiculosus* | | 0.39 | 3.5 | [63] |
| *Sargassum vulgare* | | 0.09 | 3.5 | [63] |
| *Sargassum sp* | | 0.61 | 5.5 | [33] |
| *Padina* sp. | Ni(II) | 0.63 | 5.5 | [33] |
| *Cystoseria indica* | | 0.85 | 6.0 | [70] |
| *Nizmuddinia zanardini* | | 0.94 | 6.0 | [70] |
| *Sargassum glaucescensand* | | 0.94 | 6.0 | [70] |
| *Padina australis* | | 0.46 | 6.0 | [70] |
| *Fucus spiralis* | | 0.85 | 6.0 | [46] |
| *Ascophyllum nodosum* | | 0.73 | 6.0 | [46] |
| *Sargassum filipendula* | | 1.07 | 4.5 | [65] |
| *Fucus vesiculosus* | | 1.21 (Cr(III)) 0.82 (Cr(VI)) | 4.5 (Cr(III)) 2 (Cr(VI)) | [62] |
| *Fucus spiralis* | Cr | 1.17 (Cr(III)) 0.68 (Cr(VI)) | 4.5 (Cr(III)) 2 (Cr(VI)) | [62] |
| *Sargassum* sp. | | 0.60 (Cr(VI)) | 2 (Cr(VI)) | [71] |
| *Sargassum muticum* | | 3.77 (Cr(VI)) | 2 (Cr(VI)) | [72] |

a = Not maximum biosorption value.

Table 3 shows the different species of algae used in the removal of heavy metals. The numbers for metal ion uptake qmax (mmol/g) for the different species are in the range (0–4), especially the brown alga species (*Sargassum muticum*), while all uptake occurs between pH values of (2–6), and pH influences the dissociation of heavy metals from the solution using different alga species [48,73]. The pH impacts metal ion uptake, which is a result of the influence of the "functional group on the biomass' cell wall and the metal ions solution" [33]. The polysaccharides present in the cell wall of seaweeds are the most highly metal-binding sites [64].

### 3.2. Various Natural Materials Used for Sorption

In recent years, engineers and scientists have directed much effort towards identifying the most suitable biosorption materials. Among many materials, seaweed has been revealed to be the most suitable and effective natural material. Table 4 shows some of the various other materials that have been used for the removal of metal ions.

**Table 4.** Various natural materials used for the removal of metal ions.

| Materials Used | Heavy Metals | References |
|:---:|:---:|:---:|
| Polymers | Fe and Cr | [74] |
| Sawdust and tree barks | Hg, Pb, and Zn | [75] |
| Electronic waste along with galvanic wastes | Cu, Ni, Mn, Pb, Sn | [76] |
| charcoal: | Cr(III) | [77] |
| Clay | Cr(III) | [78] |
| Fungi | Cr, Fe | [79] |
| Dead biomass | Cr | [80] |
| Peat moss | Cr, Fe | [81] |
| Peanut shells, Rice husk, Straw, and walnut cover | Cr, Cu, Ni | [82] |
| Cocoa shell | Al, Cd, Co, Cr, Cu, Fe, Mn, Ni, Pb, and Zn | [83] |
| Coconut husk | Cr, As | [82] |
| Caol and fly ashes | Cr, Cu, Ni | [84] |
| Banana pith and peels | Ni, Pb | [85] |
| Cassava fiber | Pb, Co | [86] |
| Chicken feathers | Al, As | [87] |
| Sheep manure wastes | Ca, Cd | [88] |
| Sunflower | Co, Cr | [89] |
| Rice byproducts | Cu, Fe | [90] |
| Orange peels | Cu, Fe, Hg | [91] |
| Palm kernel fiber | Fe, Hg | [82] |
| Grape stalks | Cr, Fe, Hg | [92] |

As highlighted in Table 4, the use of different biomass (living or dead) for the removal of heavy metals has been studied over the years, and microalgae have stood out among the others. For non-living organisms, the cell surface involves different functional groups like amini, hydroxyl, sulfhydryl, phosphate, sulfate, and carboxyl groups [93]. Sawdust and tree barks are rich in tannin/lignin, and have been studied by Fiset and team [94], as they proved effective in metal adsorption. The tannin is an active species during the metal adsorption (ion exchange) process because of the polyhydroxy polyphenol groups [95]. Lignin, which is extracted from black liquor and is also a waste product of the paper industry, has been considered for the removal of metals (Hg, Pb, and Zn) [96]. Alcohols, acids, aldehydes, ketones, phenol, hydroxides, and ethers are all polar functional groups of lignin that have varying metal-binding capabilities [97]. Phytoremediation or phytofiltration of metal-contaminated effluents have been tested and proven successful. Some examples of aquatic plants with such ability are *Ceratophyllum demersum*, *Lemna minor, and Myriophyllum spicatum* [98]. Cellular components such as amide, imine, imidazol moieties, carboxyl, hydroxyl, sulfate, sulfhydryl, phosphate of these plants have high metal-binding properties, as reported by Gardea and team [99]. Chitin and chitosan have also been used to treat metal ions in wastewater. Chitin, which is the second-most abundant natural biopolymer after cellulose, is commonly found in the exoskeletons of crustaceans and shellfish, while Chitosan is produced by alkaline N-deacetylation of chitin [100]. Similarly, peat moss has been studied based on heavy metal decontamination of wastewater. It is a complex material with both lignin and cellulose as its main constituents, which contain polar functional groups [101]. Plenty of other agricultural waste, such as rice residues, fruit

and vegetable peels, tea/coffee residues, and coconut husks, have also been used for metal ion retention. Most of the materials have polyhydroxy, polyphenol, carboxylic, and amino groups, which play key roles in the metal adsorption process [83]. Animal bones, clay, human hair, and teeth have all been used to treat metal ions, but have not been effective or efficient when compared with seaweed [102]. In conclusion, the above-discussed natural sorption materials have not been effective either in terms of metal ions removal rate or socio-economic benefit when compared to seaweed.

## 4. Sorption Mechanism of Seaweed

Seaweed is characterized by both physical, biological, and chemical attributes, such as alginate, carrageenan, and photosynthesis features. It can also grow in extreme conditions, in the presence of heavy metals, salinity, and harsh temperatures. Owing to the aforementioned qualities, in addition to its high binding affinity, seaweed is considered a good bioremediation material for treating toxic metal ions in aqueous solutions [103]. Seaweed also has a "hormesis phenomenon feature", which refers to the toxic contamination of algae stimulating further algae growth [104]. Similarly, some cyanobacteria tend to grow in wastewater that is highly polluted with toxic heavy metals; examples of cyanobacteria include; spirogyra, oscillatoria, anabaena, and phormidium [105]. Seaweeds have both antioxidant enzymes and non-enzymatic antioxidants. Antioxidant enzymes include catalase, superoxide dismutase (SOD), ascorbate peroxidase, and reductase, while non-enzymatic antioxidants include glutathione (GHS), cysteine, proline, carotenoids, and ascorbic acid (ASC) [106]. During the sorption process, heavy metals in the seaweed ignite the phytochelatins (PCs) through biosynthesis. These phytochelatins are proteins and thiol-rich peptides that can minimize toxic metal ions through interaction [107]. Superoxide dismutase (SOD) performs a defensive role against the superoxide anion, which is exerted by breaking the superoxide anion into hydrogen peroxide and oxygen molecules. The catalase degrades hydrogen peroxide to oxygen and water molecules, while cysteine is the precursor for metallothioneins, phytochelatins (PCs), glutathione (GSH), and other sulfur-related compounds. [108]. The reduction of free radicals and reactive oxygen species (ROS) is performed by both glutathione (GSH) and ascorbic acid (ASC), which are endogenous antioxidants that are synthesized by seaweed [109]. Additionally, seaweed produces a high level of ascorbic acid (ASC) as "hydrophilic redox buffer", which protects cytosol against the threat of oxidation. Similarly, the seaweed is protected by glutathione (GSH) by enabling phytochelatins (PCs), scavenging free radicals, and ascorbic acid (ASC) synthesis alongside the restoration of substrate for other antioxidants [106,107]. The chemistry involved in the interaction between the biomass (seaweed) and the metal ions is shown in Figures 5 and 6, respectively.

As shown in Figure 5, the removal mechanism of heavy metals is performed in two folds. These two folds include biosorption, which is the "rapid extracellular passive adsorption", and the latter is bioaccumulation, which is the "slow intracellular positive diffusion and accumulation". Seaweeds' cell walls are made up of cellulose and alginate (polysaccharides) and lipids, while the organic protein offers amino, phosphate, hydroxyl, thiol-rich, and carboxyl (functional groups), which all possess good ability to bind metal ions [105]. Additionally, the cell wall is composed of laminarin, deprotonated sulphate, and monomeric alcohols capable of attracting both cationic and anionic species of metal ions [110]. Adsorption on the surface of seaweed occurs rapidly when compared to inside the seaweed. On the surface, adsorption takes place through ion exchange with the cell wall and covalent bonding with the ionized cell wall, resulting in "seaweed exopolysaccharides". Conversely, adsorption is slow inside, and phytochelatins, GSH, and metal transporter play a leading role in the binding of metal ions. This accumulation of metal ions inside is carried across the cell membrane to the cytoplasm before diffusion [110,111].

According to Figure 6, the biochemical constituent of seaweed is responsible for the sequestration of metal ions, which are composed of alginate and fucoidan in the cell wall. The cell wall of microalgae is made up of a fibrillary skeleton (cellulose) and

an amorphous embedded matrix (alginate) [5]. The cell wall of brown algae contain sulfated polysaccharides, while in red algae, galactans are found, and green algae, hydroxyl-proline [46].

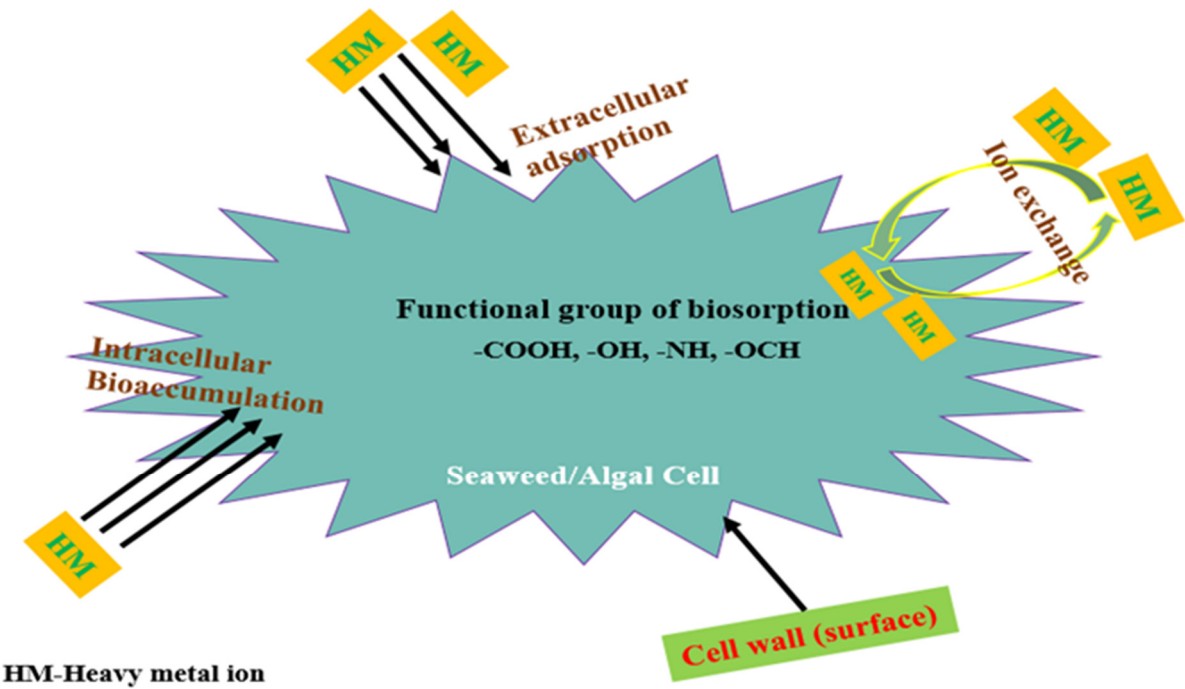

**Figure 5.** Mechanism of metal ion interaction with seaweed.

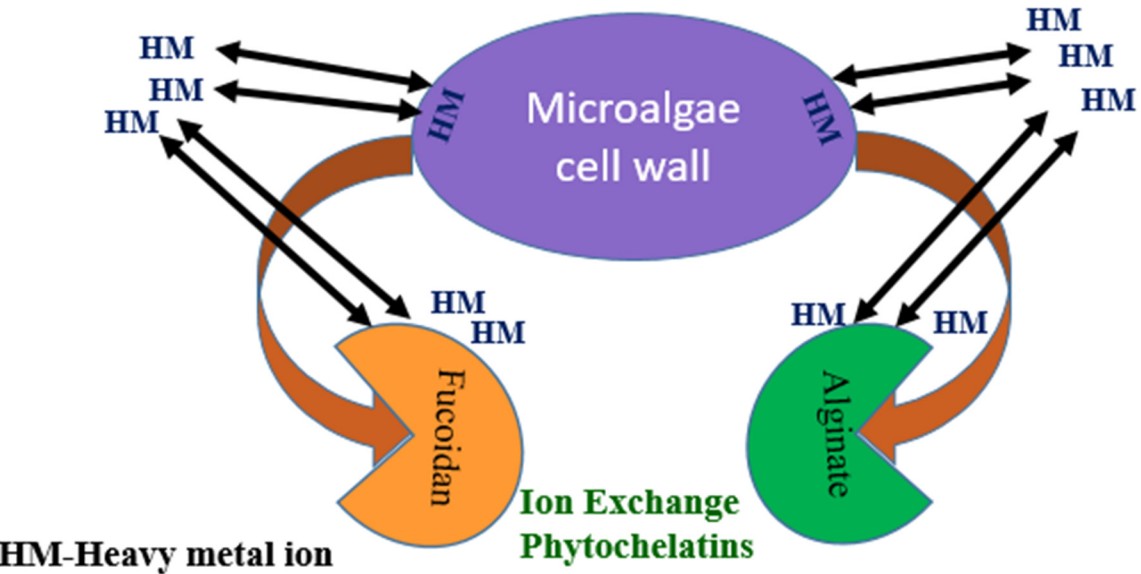

**Figure 6.** Interaction between metal ions and algal biomass.

## 5. Conclusions

The usage of seaweed as a sorption material has attracted the attention of many researchers in recent times. Seaweed's relevance is not only restricted to the treatment of heavy metals; it is a precious food that is prominent in basic balanced diets. Considering the current state of heavy metal pollution in our environment, seaweed has been proven to be an excellent, cheap, effective, abundantly available, eco-friendly, and efficient material for remediating the environment when compared to other natural sorption materials. This

multi-faceted and multi-dimensional seaweed has the potential to heal the world from various environmental menaces. It is evidence that seaweed could be economically prudent both for industrial and environmental uses. As seaweeds are among the most fascinating and resourceful species, more exploration is needed to reap the benefits of these unique species. For sorption purposes, seaweed has been proven to be a good biosorption material with high metal ion uptake (qmax (mmol/g)) within the range (0–4). The brown alga (*Sargassum muticum*) stands out efficiently at a pH value of 2 when compared to other natural sorption materials. The main biochemical interaction between the algae and the metal ions depends on the cell wall, with polysaccharides, lipids, and other organic proteins being the components that play the main roles during the sorption process. In conclusion, the sorption of metal ions using seaweed, especially brown algae, presents a solution that is more reliable, cheaper, and possesses more effective sorption ability than other natural sorption materials previously studied.

**Author Contributions:** Conceptualization—E.H.F.J., B.B.; Funding acquisition—B.B.; Methodology—E.H.F.J., B.B., X.X.; Resources—B.B.; Software—E.H.F.J.; Supervision—B.B., X.X.; Validation—X.X.; Writing—original draft—E.H.F.J.; Writing—review & editing—E.H.F.J., B.B., X.X.; Project administration—B.B. All authors have read and agreed to the published version of the manuscript.

**Funding:** This work was funded by the Natural Science Basic Research Program of Shaanxi (Program No. 2021SF-497), and the Fundamental Research Funds for Central Universities (CHD 300102291403).

**Institutional Review Board Statement:** Not applicable.

**Informed Consent Statement:** Not applicable.

**Data Availability Statement:** Not applicable.

**Conflicts of Interest:** This work has no conflict of interest.

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
