# Peer review of "Removal of Toxic Heavy Metals from Contaminated Aqueous Solutions Using Seaweeds: A Review"

_sustainability, doi:10.3390/su132112311_

Round 1
Reviewer 1 Report
The article is suitable for publication in its current form.
Author Response
I have done the necessary corrections
Reviewer 2 Report
The authors reviewed the potential of seaweeds in Removal of toxic heavy metals from contaminated aqueous solutions. The topic is interesting. However, the manuscript was badly written. I don’t think it meets the standard of the journal. The main drawbacks are below:
- Section 2.0. “”Heavy Metals and sources” including "2.1. Effect of Heavy Metal on the environment" is not necessary and should be removed. Heavy metals source, toxicity and effect are widely known and well summarized. Here, you did not provide new findings or opinions. Please focus on your topic “seaweeds to remove metals”. I suggest a brief introduction on heavy metal contamination, especially water.
- Sometimes, the logic is difficult to follow. For example, in L21, Just “as it has become a universal issue and attracted global attention”, we should over-emphasize it.
- The manuscript is wordy and should rephrase. For example, L180-18 and L235-236 are meaningless.
- The language does not meet the standard. I strongly suggest the authors improve the language. Some examples:
L23: “man and his”, this is awkward use, rewrite;
L27: How do you know they are “ions” but not other forms?
L36: replace metals with metal;
L42: eco-friendliness;
L157: What does “three (3)” mean?
Reviewer 3 Report
The paper is an interesting review on the application of seaweeds in metal removal. Before acceptance for publication it needs a fundamental revision throughout the paper: it must be explained if the review deals with seaweeds used as living cell in a bioremediation system, or it is used as dead material in a cartridge or filetr plant, or both of them. It is not clear and this is a fundamental point of the review.
Add page and line numbers for reviewing!
Other amendments:
1) Confirm that you have the authorisations and rights to reproduce figures 2,3 and 4
2) Table 1: put the right reference for EU standard which is the Directive (EU) 2020/2184 of the European Parliament and of the Council of 16 December 2020 on the quality of water intended for human consumption (recast)
3) Table 2: Row: Mercury, second column: is it "leads" as verb, or "Lead" as Pb? I think you made some mistake with automatic substitution
4) Table 3 , fourth column: common names shouldn't be in italics, italic letters are only for scientific names
5) last page before references: sixth line: I do not understand the meaning of the word "adept". Is it "adapted"? or something else?
5) last page, first line of conclusion: why "MY researchers"? Whose are the researchers?
Round 2
Reviewer 2 Report
I don't think the authors have made substantial revisions.
Reviewer 3 Report
As regards table 1, I recommend to insert the original references to the official documents on standards and limits for every organisation (not only EU), and not cite them as related references from other papers. It is more correct to consult the original documents and reports from each institution, and verify the current status of the legislation and check data.
As an example, data for EU Directive 2021/2184 should be updated and amended in Table 1: Pb is 5 ug/L and not 10 ugL and Cr is 25 ug/L and not 50 ug/L. Mn and Fe are only indicator parameters and not chemical parameters to monitor.
More scrupulousness is needed when writing reviews, check data on the original documents, not take them "de relato"
